# Salinity Stress Alters the Secondary Metabolic Profile of *M. sativa*, *M. arborea* and Their Hybrid (Alborea)

**DOI:** 10.3390/ijms22094882

**Published:** 2021-05-05

**Authors:** Efi Sarri, Aikaterini Termentzi, Eleni M. Abraham, George K. Papadopoulos, Eirini Baira, Kyriaki Machera, Vassilis Loukas, Fotios Komaitis, Eleni Tani

**Affiliations:** 1Department of Crop Science, Laboratory of Plant Breeding and Biometry, Agricultural University of Athens, Iera Odos 75, 11855 Athens, Greece; sarri@aua.gr (E.S.); gpapadop@aua.gr (G.K.P.); vassilis.loukas@gmail.com (V.L.); 2Laboratory of Pesticides’ Toxicology, Department of Pesticides Control and Phytopharmacy, Benaki Phytopathological Institute, 8 St. Delta Street, Kifissia, 14561 Athens, Greece; a.termentzi@bpi.gr (A.T.); e.baira@bpi.gr (E.B.); k.machera@bpi.gr (K.M.); 3Faculty of Agriculture, Forestry and Natural Environment, School of Forestry and Natural Environment, Aristotle University of Thessaloniki, 54124 Thessaloniki, Greece; eabraham@for.auth.gr; 4Department of Biotechnology, Laboratory of Molecular Biology, Agricultural University of Athens, Iera Odos 75, 11855 Athens, Greece; fotiskomaitis@hotmail.com

**Keywords:** salinity tolerance, secondary metabolites, *Medicago sativa*, *Medicago arborea*, Alborea, abiotic stress

## Abstract

Increased soil salinity, and therefore accumulation of ions, is one of the major abiotic stresses of cultivated plants that negatively affect their growth and yield. Among Medicago species, only *Medicago truncatula*, which is a model plant, has been extensively studied, while research regarding salinity responses of two important forage legumes of *Medicago sativa* (*M. sativa)* and *Medicago arborea* (*M. arborea*) has been limited. In the present work, differences between *M. arborea*, *M. sativa* and their hybrid Alborea were studied regarding growth parameters and metabolomic responses. The entries were subjected to three different treatments: (1) no NaCl application (control plants), (2) continuous application of 100 mM NaCl (acute stress) and (3) gradual application of NaCl at concentrations of 50-75-150 mM by increasing NaCl concentration every 10 days. According to the results, *M. arborea* maintained steady growth in all three treatments and appeared to be more resistant to salinity. Furthermore, results clearly demonstrated that *M. arborea* presented a different metabolic profile from that of *M. sativa* and their hybrid. In general, it was found that under acute and gradual stress, *M. sativa* overexpressed saponins in the shoots while *M. arborea* overexpressed saponins in the roots, which is the part of the plant where most of the saponins are produced and overexpressed. Alborea did not perform well, as more metabolites were downregulated than upregulated when subjected to salinity stress. Finally, saponins and hydroxycinnamic acids were key players of increased salinity tolerance.

## 1. Introduction

Soil salinization is a major problem facing agriculture and contributes to reducing productivity of arable lands mainly in arid and semi-arid areas [1]. According to FAO 1997 [2], soil with an electrical conductivity of the saturation extract (ECe) more than 4 dS m^−1^ is defined as saline. The problem is steadily increasing and it is predicted that more than 50% of the arable land is going to be salinized by 2050 [3] as a consequence of the expected increase of irrigated land as well as due to the expected global climate change [4]. The effect of salinity on crops depends on species, cultivars, environmental factors and plant growth stage [5]. The loss of crop productivity in saline soil is a consequence of the negative impact of salinity on plant growth and development [6].

The depression of plant growth under high levels of NaCl concentrations is a result of the reduction of water potential (hyperosmotic stress) and the accumulation of ions to a toxic level for their growth (hyperionic stress) [7]. Plants respond to these stresses by modifying their physiological and biochemical processes. Thus, researchers have focused on plants’ signaling perception and adaptation to an adverse environment by monitoring gene regulation, expression of functional proteins and accumulation of small molecules (i.e., metabolites) [5,8,9]. Several studies have implicated the role of primary metabolites such as sugars, amino acids and organic compounds in osmotic regulation. On the other hand, secondary metabolites being the end products of primary metabolites are more species specific and are associated with plant protection by having multiple functions (e.g., acting as antioxidants, reactive oxygen species (ROS) scavengers, regulatory molecules) [10]. Although the synthesis of secondary metabolites such as phenols, saponins, flavonoids, carotenoids, lignins, etc. generally increases when plants are cultivated under salt stress conditions [11], special attention has been given only to afew target compounds in previous studies [12,13]. Thus, the association of antioxidant activity and accumulation of phenolic compounds under salt stress has been confirmed by several studies [14,15]. Other secondary metabolites which were found in high abundance and are possibly involved in salt tolerance by rendering a more efficient antioxidant capacity are flavonoids [16], polyphenols, tannins, and anthocyanins [17]. In a recent work, Yang et al. (2017) [18] used saponin as a priming agent to improve the seed germination of quinoa plants in saline conditions due to the fact that saponins can act as ROS scavengers. In this regard, by analyzing the changes of metabolites at the whole-metabolome scale [19] and by using these metabolic profile changes in combination with other ‘omic’ analyses, such as genome, transcriptome and proteome analysis, one can elucidate the regulatory networks and identify biomarkers that regulate stress responses and are useful in plant improvement [20,21].

*Medicago sativa* L. (lucerne, alfalfa) (*M. sativa*) is a perennial legume of high importance in animal feeding. It is the main fodder crop in many countries worldwide for its high nutritive value as well as its contribution to soil fertility due to nitrogen fixation. Beyond its use as animal feed, *M. sativa* is also important as cover crop [22] and as a source of biologically active compounds (secondary metabolites) for the agriculture and pharmaceutical industries [23]. On the other hand, *Medicago arborea* L. (tree medick) (*M. arborea*) is an evergreen shrub that mainly spreads in the Mediterranean basin and it is used in animal feeding mostly during summer and winter. It is considered the oldest species of the genus and is more tolerant to abiotic stresses compared to alfalfa [24,25]. The Alborea hybrid was created by crossing the species *M. sativa* and *M. arborea* in order to meet the great needs for animal feed and high biomass production as well as resilience to environmental changes [26].

There are only a few studies on the metabolite profile of the Medicagos species including the model legume *M. truncatula*. In particular, Barsch et al. (2006) [27] determined the characteristic metabolites for *M. truncatula* root nodules. Recently, pathway-specific metabolome analysis using 18O2-labeled *M. truncatula* was reported by Kera et al. (2018) [28], while Dickinson et al. (2018) [29] combined transcriptomic techniques and k-means clustering in metabolomics to detect biotic and abiotic stress markers for *M. truncatula*. There is only one work regarding the changes of primary metabolites in *M. sativa* when subjected to salt stress treatment [30]. To our knowledge, there is no research regarding the modification of secondary metabolite profiles under salt stress of Medicagos. In this regard, the aim of the present study was the identification of the secondary metabolite profile in shoots and roots of *M. sativa*, *M. arborea* and their hybrid Alborea under salinity conditions, after application of two salt treatments: (a) continuous application (acute stress) of 100 mM NaCl, and (b) a gradual increase in salt concentration up to 100 mM.

## 2. Results

### 2.1. Growth Parameters and Salinity Sensitivity Index

The studied growth parameters of seedlings (stem height, stem elongation rate, salinity sensitivity index) were significantly affected by salt treatments, entries and dates of treatments (Table 1). The only exception was the stem height and the stem elongation rate (SER) that were not affected by the entries. In all cases, the interaction between salt treatments and entries was significant, indicating different responses of the entries to salt treatments (Table 1). The stem height and SER of the hybrid (Al) were significantly reduced under both salt stress of 50-75-100 mM and salt shock of 100 mM (Figure 1). Regarding the parental species, the stem height and SER of *M. sativa* were significantly reduced only under salt shock of 100 mM (Figure 1). On the other hand, the stem height and SER of *M. arborea* tended to decrease under both salt stress and shock, but this did not show statistical differences (Figure 1). It has to be mentioned that the stem height and SER of the Alborea population were the highest under control while both parameters did not significantly differ among the entries under salt stress. However, the population of *M. arborea* had the highest stem height and SER under salt shock (Figure 1). As a result, the highest salinity sensitivity index (absolute number) was recorded for the population of Alborea, while the lowest was recorded under salt stress for the population of *M. sativa* and under salt shock for the population of *M. arborea* (Figure 2).

### 2.2. Metabolomics Analysis

#### 2.2.1. Comparison among the Entries

Regarding the secondary metabolic profiling, PCA analysis clearly demonstrated that *M. arborea* differed from the other two entries (*M. sativa* and Alborea) by comparing either all treatments or each individual treatment between entries in both shoots and roots (Figure 3, R1-S3). On the other hand, the separation of the Alborea and *M. sativa* samples in all categories compared was much more blurred and in some cases the permutation tests showed no differentiation of the two groups. We drew the same conclusions by conducting PLS-DA. Permutation tests, allowing 100 permutations, indicated good predictability and goodness offit for PLS-DA models in all cases (details are omitted).

Specifically, for all treatments (Figure 3, R1 & S1), it was clearly shown that the PC1 axis, which is a linear combination of secondary metabolites for all treatments and interprets 38.3% and 36.5%, respectively, of the total variability of the phenomenon, differentiated the species as follows: in one group *M. arborea* and in another group *M. sativa* and Alborea. The metabolites that differentiated *M. arborea* from the other entries were mainly saponins (e.g., medicagenic acid, caryophyllogenic type), (Appendix A).

However, at 100 mM NaCl (Figure 3, R2, S2) and at 50-75-100 mM NaCl a tendency to distinguish the metabolic profile of *M. sativa* from Alborea was evident. More specifically, for the gradual stress treatment of 50-75-100 mM NaCl for the shoots, phenolic compounds and some triterpenic saponins were highly induced for all species. Nevertheless, for *M. arborea*, hydroxycinnamic acids, a class of phenolic compounds, were overexpressed. Moreover, some flavonoids and saponins were upregulated as well. On the other hand, *M. sativa* and Alborea upregulated mainly different triterpenic saponins and flavonoids, compared to *M. arborea*, (Appendix A). For the roots, under gradual salinity stress, the VIPs for *M. arborea* were exclusively saponins (mainly triterpenic) while for *M. sativa* and Alborea, apart from saponins, some phenols and flavonoids were also detected (Table 2). It is noteworthy that the secondary metabolite that mostly characterized the Alborea roots extracts under gradual salinity was tentatively identified as a diphenyl, most probably a lignan of eudesmin type, and was absent from *M. arborea*.

In general, under acute stress, the VIPs at the shoots for the *M. arborea* metabolomic profile were like those of gradual salinity stress, which implies a similar way of reaction. In particular, *M. arborea* plants were still the only ones that produced hydroxycinnamic acids (caffeic, ferulic, cinnamic acids). Those included mainly flavonoid glycosides. However, under acute stress, *M. sativa* significantly differed from Alborea by overexpressing some unique compounds such as a flavonoid compound, probably an apigenin derivative, and two saponins. On the other hand, Alborea overexpressed a special category of phenols, a lignan, that was also abundant in the roots (Table 3).

Concerning the metabolites accumulated in roots under acute stress, in *M. arborea* many triterpenic saponins, that are species specific, mainly glycosides of the aglycons zahnic and medicagenic acids, were present. On the other hand, in *M. sativa* roots under acute stress, no significant accumulation of secondary metabolites was detected, whereas Alborea reacted under acute stress by increasing its characteristic lignans that have been previously mentioned (Table 3). In general, it seems that under acute stress, *M. sativa* overexpressed saponins in the shoots while *M. arborea* overexpressed saponins in the roots. Alborea, on the other hand, displayed an increased production of lignans and phenyl tetrahydrofurans mainly in the roots. 

As mentioned above, the Alborea and *M. sativa* species roughly differed when it came to their secondary metabolite content in both shoots and roots. The permutation tests showed only small differentiations presented in Appendix A. The fact that *M. arborea* expressed different metabolites compared to the other two entries was also shown by the heat maps (Figure 4 and Figure 5). Thus, the metabolites that were upregulated in *M. arborea* (in blue), were downregulated in the other two entries (in red) and vice versa. 

Specifically, for Figure 4 (roots), it was evident that *M. arborea* in both stressed and control plants up- or downregulated roughly the same group of metabolites. For Alborea and *M. sativa* no clear trend was detected. However, for *M. sativa,* more metabolites were accumulated in gradual stress than in acute stress compared to control. The same results are captured in Figure 5.

#### 2.2.2. Comparison between Roots and Shoots

A distinct set of secondary metabolites was expressed in the roots compared to shoots (Figure 6). More specifically, for the *M. arborea* and Alborea, a clear difference between the metabolites produced in the shoots compared to roots was demonstrated, regardless of the treatment (Figure 6a,c). For *M. sativa*, however, in addition to a clear separation of metabolites between roots and shoots, a differentiation of secondary metabolites was apparent, even for each treatment separately (Figure 6b).

From the comparison of the Table 2a,b and Table 3a,b, it is obvious that roots and shoots of the same treatments produce different secondary metabolites that characterize salinity response. In general, roots are the parts of the plants where most of the saponins are produced and overexpressed, especially under salinity stress. Specifically, for *M. arborea* species, saponins were almost exclusively found in roots and this profile is even more pronounced under the salinity treatment. On the other hand, in the shoots other types of compounds, mainly phenols, were produced. Hydroxycinnaminc acids were exclusively found in the shoots. Flavonoid content was also more intense in the shoot extracts. Lignans on the other hand were equally characterized in the Alborea extracts in shoots and roots. Saponins in the aerial parts were usually less intensely expressed as peaks in the extract and sometimes they differed slightly, mainly in the glycosylation level.

### 2.3. Secondary Metabolomic Changes in Response to Salt Treatment

As shown in Figure 7 and Appendix A, there were differences in the expression of secondary metabolites, in some cases, between control plants and plants subjected to salinity stress for each entry. In roots of *M. arborea*, a reduction in some saponins, flavonoids and triterpenic acids in plants subjected to 100 mM NaCl and 50-75-100 mM NaCl, compared to control plants, was detected. On the other hand, the expressions of benzyl tetrahydrofurans, lignans and phenols were significantly higher in the plants under gradual stress compared to the controls. For the other treatments, there were no significant differences in roots or shoots. According to our results, *M. arborea* altered the accumulation of secondary metabolites when subjected to salinity stress, only in roots and not in shoots.

For *M. sativa*, flavonoids and phenols were decreased compared to control plants, while lignans, benzyl tetrahydrofurans and phenols were increased in roots. In the shoots, biphenyl and flavonoids were decreased, while benzyl tetrahydrofuran, phenol and lignans were increased in plants receiving 50-75-100 mM NaCl compared to control plants.

Finally, Alborea plants, which were subjected to gradual stress, showed higher levels of a specific flavonoid in roots compared to control plants. On the other hand, the comparison between control plants and plants subjected to acute stress showed reduced levels for saponins and triterpenic acids while an increase was observed for benzyl tetrahydrofuran and lignans for plants receiving 100 mM NaCl. Regarding the shoots, there was an overexpression of benzyl tetrahydrofuran, triterpenic acid, flavonoids and lignans in the plants that were subjected to acute stress compared to the control plants, while on the contrary there was a decrease in phenol accumulation in the plants that received 100 mM NaCl compared to control. For the shoots of Alborea plants that were subjected to gradual stress of 50-75-100 mM NaCl, overexpression of benzyl tetrahydrofurans and lignans were observed in relation to the control plants.

## 3. Discussion

The accumulation of ions, due to increased salinity in the soil, is one of the major abiotic stresses of cultivated plants that negatively affect their productivity. Among Medicago species, the responses to salt stress of *Medicago truncatula*, which is a model plant, and *Medicago sativa* L., have been extensively studied [31,32] and to a lesser extent so have the responses of *Medicago arborea* [25,33,34]. Nevertheless, research of the metabolomic profile of the Medicago species upon salt stress has been very limited. In the present study the comparative alterations of the secondary metabolomic profile in roots and shoots of *M. sativa, M. arborea* and their hybrid Alborea were investigated under acute salinity stress of 100 mM NaCl and gradual increase in salt concentration up to 100 mM NaCl.

According to the results of the present study, the hybrid in terms of seedling growth performed better compared to parental species under well-growing conditions but it was more sensitive under salinity stress. On the other hand, regarding the seedling growth of the parental species, *M. sativa* and *M. arborea* were not affected by the gradual application of NaCl but reduced about 20% and 10%, respectively, in their growth under acute salinity stress. These results are consistent with the results of our previous experiment [25] as well as the results of a preliminary experiment in order to confirm their performance under salt conditions.

The secondary metabolic profiles of *M. arborea* in shoots and roots were clearly differentiated from that of *M. sativa* and their hybrid under both control and salinity stress conditions. Generally, saponins were the main metabolites that discriminated *M. arborea* from *M. sativa* and their hybrid. In particular, under salinity stress *M. arborea* exclusively accumulated in the roots saponins such as medicagenic acid and zahnic acid while in the shoots had few specific flavonoids and phenolic compounds (e.g., hydroxycinnamic acid). On the other hand, the metabolic profile of the hybrid was more similar to that of *M. sativa.* There was a clear differentiation between them mainly under acute salinity stress conditions where *M. sativa* upregulated few unique flavonoid and saponin compounds in the shoots while Alborea produced lignans in the roots. It was noteworthy that Alborea overexpressed a characteristic lignan in both shoots and roots under stress conditions while *M. sativa* did not accumulate any secondary metabolite in the roots under acute salinity stress.

Accumulation of secondary metabolites is of particular interest, as they usually regulate responses to environmental stressors, such as high and low temperatures, drought, salinity, pathogen attack etc. [35,36,37]. Very recent findings unravel the multifunctional role of secondary metabolites, either as plant growth regulators or as primary metabolites in a broad sense [36].

Among the Medicago species, the research about the secondary metabolites has mainly focused on *M. sativa* and *Μ. truncatula.* On the other hand, information on the profile of *Μ. arborea* secondary metabolites is limited. The majority of the studies emphasized saponins and flavonoids which are of highest interest because of their pharmaceutical use [38]. Saponins of Medicago species are triterpenoid compounds and more than thirty-three different saponins containing one or more sugar chain units have been identified in *M. sativa*. Among them, medicagenic acid, hederagenin, zahnic acid and soyasapogenols A and B are the main saponins of *M. sativa* aerial parts [39,40,41]. Nevertheless, the content of medicagenic acid in several cases is higher in roots than in aerial parts [42]. This was the case in the present study. The accumulation of the aforementioned metabolites was characteristic of *M. arborea* roots rather than *M. sativa* aerial parts.

Accumulation of such compounds has been associated with a plethora of biological activities, including seed germination, allelopathy, poor digestibility and beneficial anti-pathogen (including fungi and insects) properties [43]. Moreover, *M. arborea* has high allelopathic potential that affects the growth and germination of other species [44]. It is possible that this characteristic of *M. arborea* is associated with the accumulation of these compounds in its roots in the present study. Furthermore, the presence of the aforementioned metabolites and especially zahnic acid in shoots of Medicago is related tolow palatability of forages [23]. The *M. sativa* population used in the present study comes from breeding and as a consequence it contains a low amount of anti-nutritional factors in its aerial parts.

Although reports in Medicago species relate saponins to biotic stress tolerance, several reports in other species highlight their importance in abiotic stress responses [45,46]. Special attention should be given to triterpenoid compounds and their accumulation upon abiotic stresses, as they may contribute to fortification of plant membranes. Moses et al., 2014 and Pulvento et al., 2012 [43,47] studied the response of quinoa plants (a salt-tolerant species) under conditions of increased salinity and found that the plants had accumulated high levels of saponins compared to control plants. Another study that correlated saponins with plant salinity tolerance was conducted by Oku et al., 2003 [48] in which the concentration of triterpenoids increased significantly under high salinity in both shoots and roots of *Kandeliacandel* and *Bruguieragymnorrhiza.*

As previously mentioned, *Μ. arborea* changed the profile of secondary metabolites in shoots mainly by over producing hydroxycinnamic acids. The three polyphenols (caffeic acid, ferulic acid and cinnamic acid) belong to hydroxycinnamic acids (HCAs) and are all members of the biochemical pathway leading to lignin formation. Additionally, ferulic acid, apart from acting as an ROS scavenger, can positively regulate the activity of other scavenging enzymes and can also inhibit the expression of enzymes involved in ROS generation [39,40].

Regarding secondary metabolites, changes in each entry in relation to its control, *M. sativa*, deeply altered the accumulation of metabolites in both shoots and roots (mainly phenolic compounds), to face the harmful effects caused by salt stress. On the other hand, *M. arborea* showed modifications in the levels of few metabolites, mainly phenolic compounds (phenol tetrahydrofurans and lignans), only in the roots while no modification of secondary metabolites took place in shoots. Interestingly, both species upregulated these compounds under gradual NaCl application. Alborea upregulated only two groups of secondary metabolites and downregulated several other groups under acute stress and only one metabolite under gradual stress in roots (Appendix A). It seems that the root system changes in metabolomic profile are key players in salt tolerance. Roots contact the soil directly and should react immediately to avoid the detrimental effects of soil salinity [49]. When plants are subjected to salt stress, they can resist or reduce the damage caused by increasing osmotic adjustment, maintaining cell wall integrity and improving ROS scavenging [50,51,52]. Several studies have highlighted that phenolic compounds maintain high antioxidant activity and are involved in ROS detoxification and prevention from oxidative damage under salt stress [53,54,55]. More specifically, lignans are involved in lignification and cell wall synthesis and they exhibit antioxidant capacity, thus they have a positive role in stress tolerance [39,56,57]. It is known that lignans have remarkable ecological functions in plants, providing protection against herbivores and microorganisms [58]. However the accumulation of lignans in roots of Alborea did not provide the expected tolerance to salinity stress.

Finally, secondary metabolites such as phenolics and flavonoids might act as hormones that ultimately elicit an improved salt tolerance [59]. In several studies, salt tolerance was tightly linked to higher accumulation of phenolic compounds [60,61,62]. A very popular strategy to induce the production of several polyphenols is by application of stress hormones [39,43]. Given that *M. arborea’s* seedling growth was superior compared to the other two entries under acute stress it seems that its production of secondary metabolites, and in particular the production of saponins and polyphenols, had a positive effect under salt stress conditions without metabolic costs. Plants can minimize the yield costs of secondary metabolite production either by controlled recycling of the resulting compounds or by evolving the use of same biosynthetic machinery for multifunction purposes [36]. Gaining information considering the metabolomic pathways under salinity stress of both parents (and especially *M. arborea*) and the ways that they recycle or use the produced secondary metabolites not only to combat salt stress but for multiple purposes may be very valuable for genetic engineering Alborea to breed a more tolerant and high yielding offspring.

## 4. Materials and Methods

### 4.1. Plant Material

The *M. arborea* parents used to develop Alborea were originally collected from Greece. The *M. sativa* parent population used in this study was a hybrid of the two *M. sativa* parents. Two *M. sativa* parents were used in crosses of *M. sativa* × *M. arborea* to produce 27 initial Alborea hybrids [26]. The Alborea population used in this study was developed from intercrosses of 20 initial hybrids using methods described by Irwin et al. (2015) [63]. Seeds were obtained from E. Bingham, Agronomy Department, Univ. Wisconsin-Madison, USA.

### 4.2. Seed Pretreatment

Seeds of *M. sativa*, *M. arborea*, and Alborea were scarified with absolute sulfuric acid for 10 min and then rinsed thoroughly with sterile distilled water. Bleach solution (3%) was added for 1.5–2 min and then rinsed thoroughly. The scarified seeds were placed on petri dishes containing 0.5% agar at 4 °C overnight, and then transferred to 20 °C (dark) for 3–4 days.

### 4.3. Growth Conditions

From each species and their hybrid (hereafter entries), 60 seeds were planted, after pre-germination, in pots of 8.5 cm height and 10 cm in diameter. Each pot contained peat, Kronos N 50–300 mg/L, P_2_O_5_ 80–300 mg/L, K_2_O 80–300 mg/L, pH 5–6.5, salinity <1.75 g/L and one seed. Then all the pots were placed in a growth chamber at a constant temperature of 25 °C, and a photoperiod of 16 h of light and 8 h of darkness, where they were regularly watered and fertilized each week for a month during the salinity treatments. For each treatment, 20 plants were used for each species. The completely randomized block design was used (block to treatments).

### 4.4. Salt Stress Treatments

The plant material was subjected to three different treatments: (1) no NaCl application (control plants), (2) continuous application of 100 mM NaCl (acute stress) and (3) gradual application of NaCl at concentrations of 50-75-100 mM with increased concentration every 10 days.

### 4.5. Growth Characteristics Measurements

The length of the stems from the base to the tip was measured in each seedling at an interval of three days through the duration of the salt treatments. Stem elongation rate (SER) was estimated as SER = (T2 − T1)/t, where T1 and T2 represent the stem length at the beginning and at the end of a time t, respectively. Additionally, the salinity sensitivity index (IS) based on the stem length was estimated according to the formula proposed by [64], IS = (Hs − Ht)/Ht × 100, in which Hs and Ht represent the values of stem length of the salt-stressed and control plants, respectively.

### 4.6. Metabolites Extraction

The plant tissue was ground with nitrogen for the extraction of metabolites. Then, in 20 mg of plant tissue, 395 μL of methanol and 5 μL of ribitol were added. The samples were incubated for 15 min at 70 °C with continuous shaking and sealed with parafilm. Next, 200 μL of chloroform was added, followed by vortex. The samples were then maintained at 37 °C for 5 min and continued shaking. Then, 400 μL of water was added and then vortexed. Samples were centrifuged for 5 min at 13,000 rpm and 250 μL of the supernatant phase was transferred to a new Eppendorf. Finally, the samples were evaporated in nitrogen gas.

### 4.7. Metabolomic Analysis

The high-resolution mass spectrometry analyses of the extracts for the dereplication and the metabolomics experiments were performed on an UHPLC-HRMS/MS (Ultra High-Performance Liquid Chromatography–High-Resolution Mass Spectrometry) Orbitrap Q-Exactive platform (Thermo Scientific, Erlangen, Germany). The ultra-high performance liquid chromatography system was a Dionex Ultimate 3000 UHPLC system (Thermo Scientific, Germany). A Hypersil Gold UPLC C18 (2.1 × 100 mm, 1.9 μm) reversed phased column (Thermo Scientific, Germany) was used for the separations.

The mobile phase consisted of solvents A: aqueous 0.1% (*v*/*v*) formic acid and B: acetonitrile. Both formic acid and acetonitrile were purchased from Merck (Darmstadt, Germany).

Different gradient elutions were performed for positive and negative ion mode detection and after optimization of the chromatography the gradient applied was: T = 0 min, 5% B; T = 3 min, 5% B, T = 21 min, 95% B, T = 23 min, 95% B, T = 24 min, 5% B; T = 30 min, 5% B. The flow rate was 0.220 mL/min and the injection volume 5 μL. The column temperature was kept at 40 °C while the sample tray temperature was set at 10 °C.

The ionization was performed at HESI (Heated ElectroSpray Ionization), at both positive and negative modes. The conditions for the HRMS for both negative and positive ionization modes were set as follows: capillary temperature, 350 °C; spray voltage, 2.7 kV; S-lense Rf level, 50 V; sheath gas flow, 40 arb. units; aux gas flow, 5 arb. units; aux. gas heater temperature, 50°C. Analysis was performed using the Fourier transform mass spectrometry mode of the LTQ orbitrap (FTMS) in the full scan ion mode, applying a resolution of 70,000, while acquisition of the mass spectra was performed in every case using the centroid mode. The data-dependent acquisition capability was also used at 35,000 resolution, allowing for MS/MS fragmentation of the three most intense ions of every peak exceeding the predefined threshold applying a 10 s dynamic exclusion. Stepped normalized collision energy was set at 40, 60 and 100. Data acquisition and analysis was completed employing Xcalibur 2.1. The dereplication was performed by Compound Discoverer software (Thermo) where spectral libraries of natural products were used (*m*/*z* cloud, in house libraries).

The analyses of the samples followed a classical metabolomic sequence where all samples were triplicated. Quality control (QC) samples were created as pooled samples and were injected periodically, every 6 samples, throughout the run to assess instrument stability.

Following the overall chemical profiling of samples, compounds responsible for group clustering were further investigated. Variable Importance in Projection (VIPs) and coefficient values extracted from the Partial Least Squares-Discriminant Analysis (PLS-DA) revealed the most important features in positive ESI (ElectroSpray Ionization) mode, responsible for the differentiation of several comparisons that were performed.

The structural elucidation of features responsible for sample clustering was performed by comparison of the chromatographic and spectrometric features of each peak in comparison with data from literature and the standard compounds. The Orbitrap HR-UPLC-MS2 profiling was performed in both positive and negative modes. The high resolving power (70,000 at the full scan experiments and 35,000 for the MS/MS fragments) of the Q-Exactive Orbitrap analyser in correlation to the accurate mass measurements (m < 1 ppm for both full scan and MS/MS ions) assured the identification of the VIPs compounds responsible for the differentiation of the two groups with high confidence. The suggested EC (Elemental Composition) for molecular ions and MS/MS fragments, as well as the respective RDBeq (Ring Double Bond Equivalents) further assisted the safe identification process. However, due to the great similarity of the compounds found in the extracts, in several cases the structural elucidation was not feasible. Instead, a suggested annotation is given, while the general category in which the compound belongs is assigned, beyond any question.

### 4.8. Statistical Analysis

Differences between the entries for growth parameters and salinity sensitivity index were conducted using analysis of variance (ANOVA) followed by Tukey’s means post-comparison tests at 0.05 significance level. Statistical analysis was performed using IBM SPSS Statistics 23 for Windows (SPSS Inc. Chicago, IL, USA).

Spectrometric and chromatographic data were imported to Compound Discoverer 2.1 (Thermo Scientific) prior to statistical analysis. The appropriate set up for peak detection, deconvolution, deisotoping, alignment and gap filling procedures was applied. For the gap filling the signal to noise threshold was set at 1.5. The normalization was QC-based and the regression model used was linear. Blanks were excluded and the type was constant mean. The appropriate settings were also used for the composition prediction, setting the mass tolerance at 5 ppm. Principal component analysis (PCA) was used to group both species and their hybrids based on secondary metabolites and heatmaps were used to investigate expression differences of secondary metabolites.

The MS (mass spectrometry) data were also subjected to multivariate statistical analysis (PLS-DA) using SIMCA P + 11.5 software (Umetrics, Umea, Sweden). Permutation testing employing 100 random permutations was applied to validate the statistical models. The optimal number of principal components for all models resulted from the R2 and Q2 values. The variables that exhibited a VIP scoring greater than 1 were verified by t-test using a *p*-value at ≤0.05. These variables were annotated using the Compound Discoverer software in correlation with in-house libraries of natural products, applying *m/z* to lerance of 5 ppm and taking into consideration the isotopic and MS/MS fragmentation pattern. Heatmaps were constructed by using XLSTAT software.

## 5. Conclusions

According to our results the parental lines responded better compared to their hybrid under salinity stress conditions. Regarding metabolomic analysis, *M. arborea* upregulated saponins in the roots under both gradual and acute stress while in the shoots mostly phenolic compounds (hydroxycinnamic acids) were upregulated. Thus, these secondary metabolites may play a role in *M. arborea’s* stable performance under salinity stress (Appendix A). These findings are useful for future studies to decipher the complex plant metabolome in order to improve abiotic stress tolerance especially in such high-importance legumes. Lastly, future breeding efforts should focus on engineering biochemical pathways leading to the specific secondary metabolite synthesis (either phenolic compounds or saponins) specifically in the roots of Alborea plants.

## Figures and Tables

**Figure 1 ijms-22-04882-f001:**
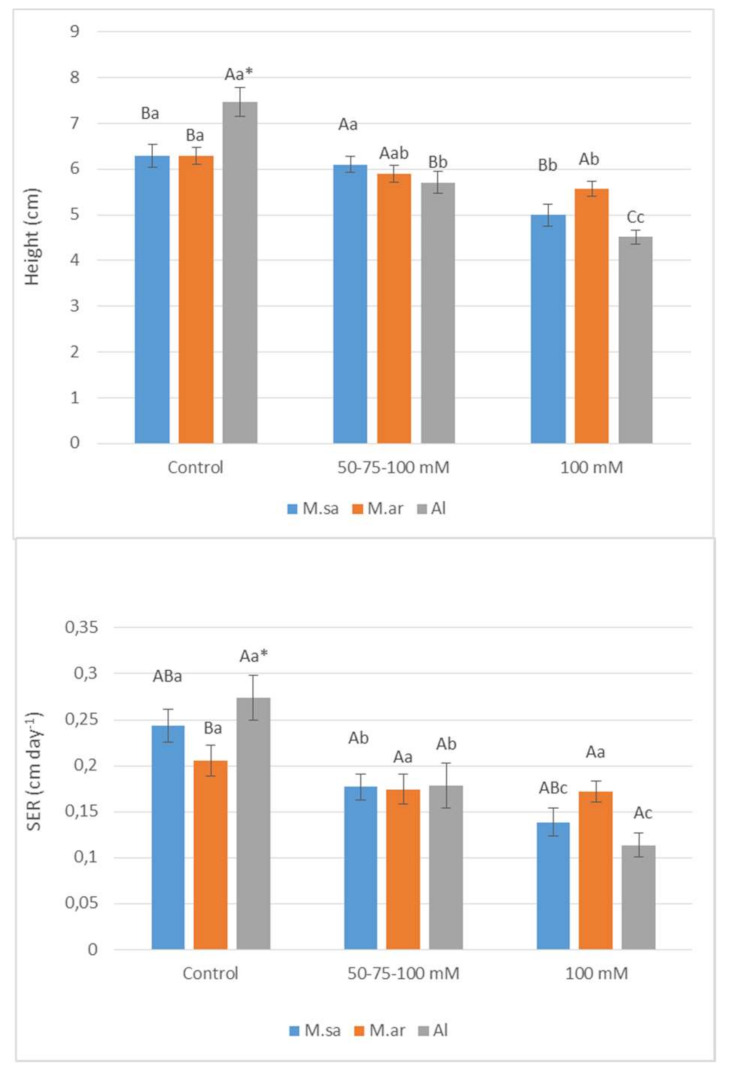
The stem height and the stem elongation rate (SER) of *M. arborea* (M. ar), *M. sativa* (M. sa), and Alborea (Al) under control, 50-75-100 mM and 100 mM. The vertical bars indicate the mean ± standard error (SE). * The different small letters indicate significant differences for the same entry among treatments; different capital letters indicate significant differences for the same treatment among the entries at *p* < 0.05 (Tukey’s test).

**Figure 2 ijms-22-04882-f002:**
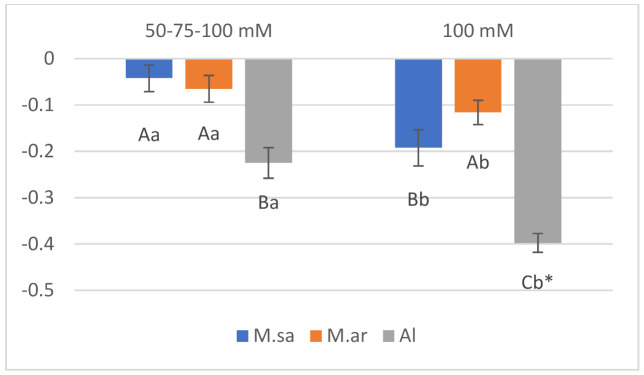
The salinity sensitivity index of *M. arborea* (M. ar), *M. sativa* (M. sa), and Alborea (Al) under 50-75-100 mM and 100 mM salt treatments. The vertical bars indicate the mean ± SE. The different letters refer to the significant differences at *p* < 0.05 (Tukey’s test). * The different small letters refer to the significant differences for the same entry among treatments; different capital letters indicate significant differences for the same treatment among the entries at *p* < 0.05 (Tukey’s test).

**Figure 3 ijms-22-04882-f003:**
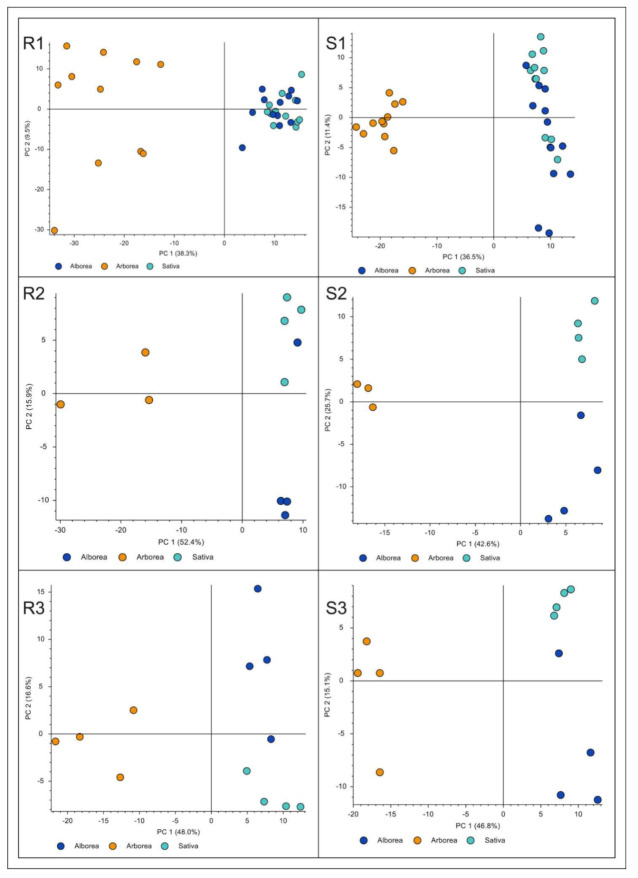
Principal component analysis (PCA) score plots of secondary metabolic profiles in Medicago entries: (R1 & S1) PCA score plots of the average for all the treatments, i.e., control, 100 mM, 50-75-100 mM, in roots and shoots, (R2 & S2) PCA score plot for, 100 mM in roots and shoots, (R3 & S3) PCA score plot for 50-75-100 mM in roots and shoots. *M. sativa* (light blue), *M. arborea* (yellow) and Alborea (dark blue).

**Figure 4 ijms-22-04882-f004:**
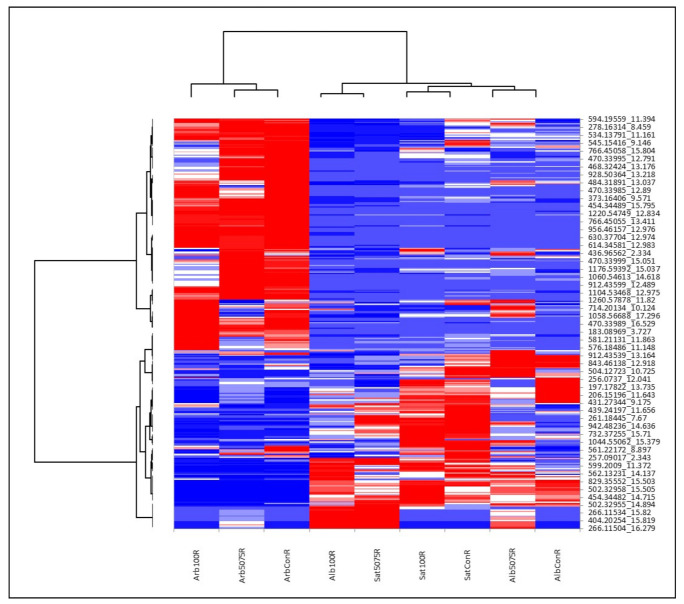
Heat map analysis of UHPLC-HRMS/MS metabolomics in roots. The row represents each metabolite and the column represents each treatment. Metabolites significantly decreased are displayed in red, while metabolites significantly increased are displayed in blue.

**Figure 5 ijms-22-04882-f005:**
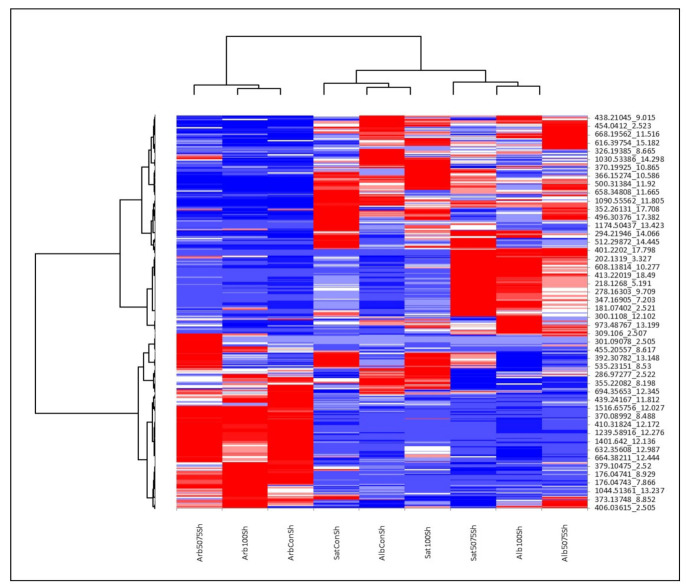
Heat map analysis of UHPLC-HRMS/MS metabolomics in shoots. The row represents each metabolite, and the column represent each treatment. Metabolites significantly decreased are displayed in red, while metabolites significantly increased are displayed in blue.

**Figure 6 ijms-22-04882-f006:**
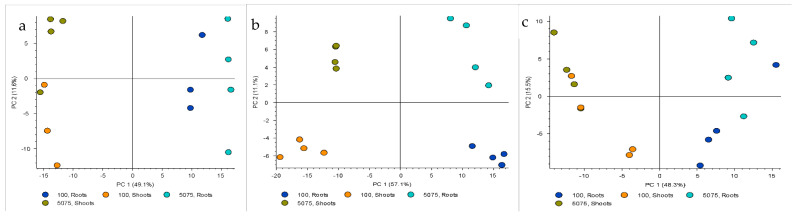
Principal component analysis (PCA) score plots of secondary metabolic profiles in roots and shoots for (**a**) *M. arborea* (M. ar), (**b**) *M. sativa* (M. sa) and (**c**) Alborea (Al): (**a**–**c**). Shoots (yellow) and roots (dark blue).

**Figure 7 ijms-22-04882-f007:**
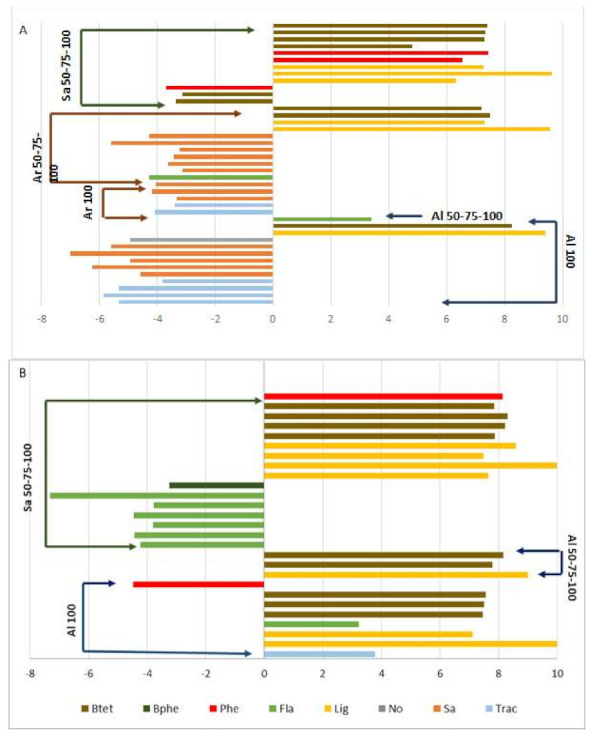
Relative fold changes of differential secondary metabolites for (**A**) roots and (**B**) shoots of *M. sativa* (Sa), *M. arborea* (Ar) and Alborea (Al) under salt stress and salt shock treatments. Btet = benzyl tetrahydrofuran, Bphe = Biphenyl, Phe = Phenol, Fla= Flavonoid, Lig = Lignan, Sa = Saponins, Trac = Triterpenic acid, No = No name.

**Table 1 ijms-22-04882-t001:** Statistical significance of F ratios from the analysis of variance for stem height of seedlings, stem elongation rate (SER) and sensitivity index.

Source of Variation	Height	SER	Salinity Sensitivity Index
Salt (A)	*p* ˂ 0.05	*p* ˂ 0.05	*p* ˂ 0.05
Species (B)	ns	ns	*p* ˂ 0.05
Dates (C)	*p* ˂ 0.05	*p* ˂ 0.05	*p* ˂ 0.05
AXB (Interaction)	*p* ˂ 0.05	*p* ˂ 0.05	*p* ˂ 0.05
AXC (Interaction)	*p* ˂ 0.05	ns	ns
BXC (Interaction)	ns	*p* ˂ 0.05	*p* ˂ 0.05
AXBXC (Interaction)	ns *	*p* ˂ 0.05	ns

* ns: not significant at 0.05 level.

**Table 2 ijms-22-04882-t002:** VIP features (secondary metabolites) with their retention times (RT), molecular weight, molecular formulas, their tentative identification and their statistical measurements that characterize each species subjected to gradual salinity stress.

a. For Aerial Parts (Shoots)						
RT [min]	Molecular Weight	Formula	Category	Log2 Fold Change	Upregulated	VIP Score
7.78	356.0742	C_15_H_16_O_10_	Caffeic acid derivative	8.01	Arb	1.08
8.28	194.0579	C_10_H_10_O_4_	Ferulic acid	6.93	Arb	1.52
8.31	370.0899	C_16_H_18_O_10_	Ferulic acid derivative	7.8	Arb	1.07
9.39	798.1487	C_33_H_34_O_23_	Flavonoid (apigenin derivative)	10.08	Arb	7.59
9.55	828.1593	C_34_H_36_O_24_	Flavonoid	7.61	Arb	1.08
10.32	828.1746	C_38_H_36_O_21_	Flavonoid	9.36	Arb	2.39
12.12	1090.5189	C_52_H_82_O_24_	Saponin	4.82	Arb	3.80
12.89	1206.5670	C_57_H_90_O_27_	Saponin	6.34	Arb	2.07
13.44	1190.5720	C_57_H_90_O_26_	Saponin	4.77	Arb	2.07
15.05	1014.5392	C_51_H_82_O_20_	Triterpenic saponin	7.22	Arb	1.52
15.33	912.5081	C_47_H_76_O_17_	Soyasaponin	4.84	Arb	1.98
16.21	910.4925	C_47_H_74_O_17_	Triterpenic saponin	6.15	Arb	
8.50	432.1267	C_18_H_24_O_12_	Iridoid	−6.49	Sat, Alb	8.68
9.04	622.1170	C_27_H_26_O_17_	Flavonoid	−6.98	Sat, Alb	3.58
9.59	974.1958	C_43_H_42_O_26_	Flavonoid	−9.21	Alb	5.02
11.07	798.1640	C_37_H_34_O_20_	Flavonoid	−6.71	Sat, Alb	5.69
11.19	1106.5505	C_53_H_86_O_24_	Triterpenic saponin	−7.84	Sat	1.01
11.43	768.1534	C_36_H_32_O_19_	Flavonoid	−6.89	Sat	1.34
11.63	1090.5555	C_53_H_86_O_23_	Triterpenic saponin	−8.71	Sat, Alb	2.47
11.92	1414.6249	C_64_H_102_O_34_	Triterpenic saponin	−6.62	Sat, Alb	1.20
11.53	1118.5510	C_54_H_86_O_24_	Triterpenic saponin	−7.21	Alb	1.41
11.74	1546.6690	C_62_H_114_O_43_	Triterpenic saponin	−7.06	Sat, Alb	1.78
11.75	1414.6250	C_57_H_106_O_39_	Triterpenic saponin	−5.72	Alb	1.04
14.98	912.5081	C_47_H_76_O_17_	Triterpenic saponin	−7.9	Sat, Alb	1.98
15.39	1028.5190	C_51_H_80_O_21_	Triterpenic saponin	−7.32	Sat, Alb	1.29
15.79	488.2197	C_30_H_32_O_6_	Triterpenic acid	−10.57	Sat	4.04
**b. Roots**						
**RT [min]**	**Molecular Weight**	**Formula**	**Category**	**Log2 Fold Change**	**Upregulated in**	**VIP Score**
12.38	664.3829	C_36_H_56_O_11_	Phytolaccoside type triterpenic saponin	8.04	Arb	2.21
12.70	644.3829	C_36_H_56_O_11_	Phytolaccoside type triterpenic saponin	5.61	Arb	2.22
12.82	648.3879	C_36_H_56_O_10_	Triterpenic saponin	6.06	Arb	1.14
12.83	1220.5478	C_57_H_88_O_28_	Triterpenic saponin	7.89	Arb	2.18
12.83	1088.5044	C_52_H_80_O_24_	Triterpenic saponin	8.4	Arb	2.63
12.83	956.4618	C_47_H_72_O_20_	Triterpenic saponin	7.57	Arb	1.01
12.97	678.3620	C_36_H_54_O_12_	Saponin (monoglycosilated)	6.23	Arb	2.15
12.97	956.4619	C_47_H_72_O_20_	Triterpenic saponin	6.86	Arb	1.45
13.40	620.3930	C_35_H_56_O_9_	Steroidal Saponin	7.95	Arb	1.51
13.41	782.4457	C_41_H_66_O_14_	Steroidal Saponin	7.11	Arb	1.31
13.78	706.3936	C_38_H_58_O_12_	Saponin	6.97	Arb	1.14
14.32	1176.5935	C_57_H_92_O_25_	Saponin	4.03	Arb	1.66
15.33	1072.5455	C_53_H_84_O_22_	Saponin	4.55	Arb	2.67
15.84	926.4881	C_47_H_74_O_18_	Triterpenic saponin	5.38	Arb	1.65
10.72	504.1271	C_24_H_24_O_12_	Flavonoid	−6.05	Alb	1.15
11.18	516.0907	C_24_H_20_O_13_	Dibenzofuran	−4.7	Sat	2.10
11.54	1118.5513	C_54_H_86_O_24_	Saponin	−7.84	Sat	1.12
14.71	942.5189	C_48_H_78_O_18_	Soyasaponin I	−7.38	Sat, Alb	3.54
15.28	502.3295	C_30_H_46_O_6_	Medicagenic acid (aglycon)	−7.59	Alb	3.29
15.49	438.3132	C_29_H_42_O_3_	Phenol	−7.84	Sat, Alb	1.64
15.49	750.3823	C_39_H_58_O_14_	Saponin	−7.59	Sat, Alb	7.19
15.49	456.3240	C_29_H_44_O_4_	Triterpenic acid	−8.15	Sat, Alb	1.66
15.49	686.3662	C_38_H_54_O_11_	Triterpenic acid derivative	−7.67	Sat, Alb	1.16
15.79	386.1727	C_22_H_26_O_6_	Lignan	−11.99	Alb (vs. Sat)	13.72
16.27	1026.50357	C_51_H_78_O_21_	Steroidal saponin	−7.81	Sat, Alb	1.37

**Table 3 ijms-22-04882-t003:** VIP features (secondary metabolites) with their tetention times (RT), molecular weight, molecular formulas, their tentative identification, and their statistical measurements that characterize each species subjected to acute salinity stress.

a. For Aerial Parts (Shoots)						
RT [min]	Formula	Molecular Weight	Category	Log2 Fold Change	Upregulated	VIP Score
7.78	C_15_H_16_O_10_	356.0741	Caffeic acid derivative	8.25	Arb	1.21
8.28	C_10_H_10_O_4_	194.0579	Ferulic acid	7.84	Arb	1.65
8.31	C_16_H_18_O_10_	370.0898	Cinnamic acid derivative	7.95	Arb	1.11
9.40	C_33_H_34_O_23_	798.1483	Flavonoid (e.g., apigenin glycoside)	10.15	Arb	6.63
9.55	C_34_H_36_O_24_	828.1591	Flavonoid (e.g., diosmetin glycoside)	8.23	Arb	1.67
10.46	C_40_H_40_O_23_	888.1954	Flavonoid (flavonol glycoside)	10.21	Arb	1.73
10.68	C_39_H_38_O_22_	858.1851	Flavonoid (flavonol glycoside)	8.38	Arb	1.92
12.90	C_57_H_90_O_27_	1206.5664	Medicagenic acid saponin	7.52	Arb	1.74
8.40	C_33_H_34_O_23_	798.1492	Flavonoid (e.g., apigenin glycoside)	−7.03	Alb, Sat	5.15
8.51	C_18_H_24_O_12_	432.1268	Iridoid	−7.36	Alb, Sat	7.11
9.04	C_27_H_26_O_17_	622.1173	Flavonoid (e.g., apigenin glycoside)	−8.65	Alb, Sat	4.92
9.59	C_43_H_42_O_26_	974.1961	Flavonoid (e.g., apigenin glycoside)	−7.25	Alb, Sat	7.31
10.83	C_21_H_18_O_11_	455.0850	Flavonoid (e.g., flavone glycoside)	−5.03	Sat	1.14
11.07	C_37_H_34_O_20_	798.1641	Flavonoid (e.g., apigenin glycoside)	−5.53	Sat, Alb	5.73
11.28	C_17_H_14_O_6_	314.0790	Flavonoid	−7.69	Alb (vs. Sat)	2.21
11.63	C_53_H_86_O_23_	1090.5560	Triterpenic saponin	−6.85	Alb, Sat	2.67
11.91	C_64_H_102_O_34_	1414.6255	Saponin	−7.43	Sat	2.40
12.13	C_48_H_76_O_21_	988.4877	Triterpenic saponin	−6.89	Alb, Sat	1.24
12.48	C_69_H_110_O_37_	1530.6740	Saponin	−6.77	Sat	1.01
15.796	C_22_H_26_O_6_	386.1727	Lignan	−11.57	Alb (vs. Sat)	16.01
**b. Roots**						
**RT [min]**	**Molecular Weight**	**Formula**	**Category**	**Log2 Fold Change**	**Upregulated**	**VIP Score**
12.03	1236.5425	C_57_H_88_O_29_	Zahnic acid saponin	8.03	Arb	1.55
12.08	1104.4988	C_52_H_80_O_25_	Zahnic acid saponin	7.33	Arb	1.03
12.15	1222.5630	C_57_H_90_O_28_	Zahnic acid saponin	7.21	Arb	1.14
12.83	1220.5476	C_57_H_88_O_28_	Medicagenic acid saponin	6.81	Arb	2.13
12.83	1088.5040	C_52_H_80_O_24_	Medicagenic acid saponin	8.16	Arb	3.04
12.83	956.4616	C_47_H_72_O_20_	Medicagenic acid saponin	5.77	Arb	2.46
12.97	648.3876	C_36_H_56_O_10_	Medicagenic acid saponin	6.11	Arb	1.58
12.97	1106.5337	C_49_H_86_O_27_	Medicagenic acid saponin	5.34	Arb	1.087
12.97	648.3876	C_36_H_56_O_10_	Medicagenic acid saponin	6.48	Arb	1.36
12.97	956.4616	C_47_H_72_O_20_	Medicagenic acid saponin	5.77	Arb	2.15
12.97	632.3563	C_35_H_52_O_10_	Medicagenic acid saponin	6.73	Arb	1.04
12.97	678.3618	C_36_H_54_O_12_	Medicagenic acid saponin	5.82	Arb	2.64
12.97	824.4196	C_42_H_64_O_16_	Medicagenic acid saponin	6.74	Arb	2.22
13.41	470.3398	C_30_H_46_O_4_	Triterpenic acid	7.57	Arb	2.27
13.48		C_48_H_70_O_22_	Steroidic saponin	−8.27	Sat	−2.87
15.50	750.3825	C_39_H_58_O_14_	Steroidic saponin	−6.48	Alb	5.46
15.50	502.3296	C_30_H_46_O_6_	Triterpenic acid (e.g., medicagenic)	−7.56	Sat, Alb	−3.06
15.82	284.1258	C_14_H_20_O_6_	Lignan	−11.08	Alb	1.97
15.82	386.1727	C_22_H_26_O_6_	Lignan	−11.99	Alb	11.93
15.82	266.1152	C_14_H_18_O_6_	Simple phenolic acid	−10.19	Alb	1.85
16.50	486.3350	C_30_H_46_O_5_	Triterpenic acid (e.g., quiallic)	−5.17	Sat	1.51

## Data Availability

Data supporting reported results can be found at: https://drive.google.com/drive/folders/1j4lARpnnTZjdtA-h2ihUHqQRa2Y4VOql?usp=sharing (accessed on 4 May 2021).

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
