# Peer review of "Salinity Stress Alters the Secondary Metabolic Profile of M. sativa, M. arborea and Their Hybrid (Alborea)"

_ijms, 2021, doi:10.3390/ijms22094882_

Round 1

Reviewer 1 Report

This study presents secondary metabolite profiling in Medicago under salt stress. Although there are interesting results, I have some major concerns.

1- Extensive editing of the English language required. There are lots of grammatical and linguistic errors.

2- The normality of the data should be double-checked. Based on Figure 1, there is no cut-off between M.sa and M.ar so there is a significant difference. However, the graph shows no significant difference.

3- Figure 3 should be replaced by a high-quality one.

4- The "Discussion" part should be improved. In most parts of the discussion, the authors have repeated the results and there is no real discussion.

5- The “Conclusion” should be improved and at least discuss the importance of the results in future studies.

6- There are no line numbers. Please provide the line number in the revised version for a better review.

Author Response

Manuscript ID: ijms-1192079
Response to reviewer 1

We are grateful to the reviewers for their insightful comments on our paper. We have been able to incorporate changes to reflect most of the suggestions provided by the reviewers. Please see below, in red, for a point-by-point response to the reviewers’ comments and concerns

(x) I would not like to sign my review report

( ) I would like to sign my review report

English language and style

( ) Extensive editing of English language and style required

( ) Moderate English changes required

(x) English language and style are fine/minor spell check required

( ) I don't feel qualified to judge about the English language and style

Yes         Can be improved              Must be improved           Not applicable

Does the introduction provide sufficient background and include all relevant references?

(x)           ( )           ( )           ( )

Is the research design appropriate?

( )           ( )           (x)           ( )

Are the methods adequately described?

( )           (x)           ( )           ( )

Are the results clearly presented?

(x)           ( )           ( )           ( )

Are the conclusions supported by the results?

(x)           ( )           ( )           ( )

Comments and Suggestions for Authors

The paper is interesting, provides new insights into a crop of high economic significance and performs an orginal approach. Nevertheless, the paper has a weak point.

In figure 1  and table 1 authors show the different sensitivity of the different plants to salt stress using two parameters (Height and SER). Differences are not very convincing, and these parameters are not the standard ones to determine salt tolerance or sensitivity. So I am not sure that the pressumed tolerant or sensitive plants are as the authors state in the manuscript. Standard parameters would be fresh and dry weight, survival under salt stress conditions, or the Na/Ka ratio in leaves or roots. Also is quite indicative to provide pictures of the plants under control and stress conditions, to see which ones are more affected. I recommend to include more convincing data on the differential salt tolerance among tested species.

Authors’ response: Thank you for this suggestion. It would have been interesting to explore this aspect.

This work is the continuation of a previously published work (Tani et al. 2018) where we studied the salinity response of M. arborea, M. sativa and their hybrid Alborea. In both works both SER and salinity sensitivity index have the same trend. The hybrid was very sensitive to NaCl concentrations above 50mM whereas the two parental species showed small declines in SER between control and salt-treated plants.  Before conducting this experiment, we performed a preliminary experiment and we got results with the same trend. For this reason and since we did not have enough tissue for determination of Na/K content in all three entries (we determined the ratio only for M arborea and M sativa and this is the reason that we did not include this measurement in the manuscript), we do not have results for Na/K ratio. We were convinced about the salinity response of the three entries from the previous work. In this work we mainly focused on determining the metabolic profile of the three entries in both shoots and roots in response to salinity stress.

Reviewer 2 Report

The paper is interesting, provides new insights into a crop of high economic significance and performs an orginal approach. Nevertheless, the paper has a weak point.

In figure 1  and table 1 authors show the different sensitivity of the different plants to salt stress using two parameters (Height and SER). Differences are not very convincing, and these parameters are not the standard ones to determine salt tolerance or sensitivity. So I am not sure that the pressumed tolerant or sensitive plants are as the authors state in the manuscript. Standard parameters would be fresh and dry weight, survival under salt stress conditions, or the Na/Ka ratio in leaves or roots. Also is quite indicative to provide pictures of the plants under control and stress conditions, to see which ones are more affected. I recommend to include more convincing data on the differential salt tolerance among tested species. 

Minor points:

The meaning of the term VIP is not explained. Is an acronym? What is its meaning?

"benyl tetrahydrofurans" is this term properly written or it refers to benzyl tetrahydrofurans?

Author Response

Manuscript ID: ijms-1192079
Response to reviewer 2

We are grateful to the reviewers for their insightful comments on our paper. We have been able to incorporate changes to reflect most of the suggestions provided by the reviewers. Please see below, in red, for a point-by-point response to the reviewers’ comments and concerns

Yes         Can be improved              Must be improved           Not applicable

Does the introduction provide sufficient background and include all relevant references?

( )           (x)           ( )           ( )

Is the research design appropriate?

( )           (x)           ( )           ( )

Are the methods adequately described?

(x)           ( )           ( )           ( )

Are the results clearly presented?

( )           ( )           (x)           ( )

Are the conclusions supported by the results?

( )           ( )           (x)           ( )

Comments and Suggestions for Authors

This study presents secondary metabolite profiling in Medicago under salt stress. Although there are interesting results, I have some major concerns.

1- Extensive editing of the English language required. There are lots of grammatical and linguistic errors.

Authors’ response: Thank you for pointing this out. The manuscript has been revised by a native English speaker.

2- The normality of the data should be double-checked. Based on Figure 1, there is no cut-off between M.sa and M.ar so there is a significant difference. However, the graph shows no significant difference.

Authors’ response: You probably refer to Table 1 and Figure 1. Based on the statistical analysis there is no significant difference among the species on average however, there is a significant interaction between species and salt treatment. Moreover, we show the significant differences among the entries in a different way using capital letters for differences among the entries for the same treatment and small letters for differences of the same entry among the treatments. We think that now it is more clear the differentiation among the entries.

3- Figure 3 should be replaced by a high-quality one.

Authors’ response: We agree with the reviewer’s assessment.  Figure 3 has been replaced with a high-quality one.

4- The "Discussion" part should be improved. In most parts of the discussion, the authors have repeated the results and there is no real discussion.

Authors’ response: As suggested by the reviewer, changes have been made in the “Discussion” part.

5- The “Conclusion” should be improved and at least discuss the importance of the results in future studies.

Authors’ response: We have added the suggested content to the manuscript on “conclusions” part.

6- There are no line numbers. Please provide the line number in the revised version for a better review.

Authors’ response: Thank you for pointing this out. Line numbers are provided in the revised version of the manuscript.

Minor points:

The meaning of the term VIP is not explained. Is an acronym? What is its meaning?

Authors’ response: The term VIP is explained in paragraph 4.7, line 436.

"benyl tetrahydrofurans" is this term properly written or it refers to benzyl tetrahydrofurans?

Authors’ response: Thank you for noticing. Corrections have been made.

Round 2

Reviewer 1 Report

All my comments have been addressed. I think that this version of the manuscript can be published in IJMS.

Reviewer 2 Report

Paper can be accepted in the present form